# Schur's Positive-Definite Network: Deep Learning in the SPD cone with structure

**Can Pouliquen**
ENS de Lyon, Inria, CNRS,
Université Claude Bernard Lyon 1,
LIP, UMR 5668, 69342, Lyon cedex 07, France
`can.pouliquen@ens-lyon.fr`

**Mathurin Massias**
Inria, ENS de Lyon, CNRS,
Université Claude Bernard Lyon 1,
LIP, UMR 5668, 69342, Lyon cedex 07, France
`mathurin.massias@inria.fr`

**Titouan Vayer**
Inria, ENS de Lyon, CNRS,
Université Claude Bernard Lyon 1,
LIP, UMR 5668, 69342, Lyon cedex 07, France
`titouan.vayer@inria.fr`

## Abstract

Estimating matrices in the symmetric positive-definite (SPD) cone is of interest for many applications ranging from computer vision to graph learning. While there exist various convex optimization-based estimators, they remain limited in expressivity due to their model-based approach. The success of deep learning motivates the use of *learning-based* approaches to estimate SPD matrices with neural networks in a data-driven fashion. However, designing effective neural architectures for SPD learning is challenging, particularly when the task requires additional structural constraints, such as element-wise sparsity. Current approaches either do not ensure that the output meets all desired properties or lack expressivity. In this paper, we introduce SpodNet, a novel and generic learning module that guarantees SPD outputs and supports additional structural constraints. Notably, it solves the challenging task of learning jointly SPD and sparse matrices. Our experiments illustrate the versatility and relevance of SpodNet layers for such applications.

## 1 Introduction

The estimation of symmetric positive-definite (SPD) matrices is a major area of research, due to their crucial role in various fields such as optimal transport (Bonet et al., 2023), graph theory (Lauritzen, 1996), computer vision (Nguyen et al., 2019) or finance (Ledoit and Wolf, 2003). While various statistical estimators, i.e. *model-based*, have been developed for specific tasks (Ledoit and Wolf, 2004; Banerjee et al., 2008; Cai et al., 2011), recent advancements focus on applying generic *learning-based* approaches to estimate appropriate SPD matrices with neural networks in a data-driven fashion (Huang and Van Gool, 2017; Gao et al., 2020).

Training neural networks while enforcing non-trivial structural constraints such as positive-definiteness is a difficult task. There have been many efforts in this direction in recent years, often in an ad-hoc manner and each with their own shortcomings (see Section 2 for more details). Building on the seminal work of Gregor and LeCun (2010) in sparse coding, a promising research direction involves designing neural networks architectures from the unrolling of an optimization algorithm (Chen and Pock, 2016; Monga et al., 2021; Chen et al., 2022; Shlezinger et al., 2023). In the case of SPD matrices, algorithm unrolling presents several challenges. First, algorithms operating in the SPD cone usually rely on heavy operations such as retractions (Boumal, 2023), SVD or line search (Rolfs et al., 2012). These operations do not integrate well into a neural network architecture, making these algorithms difficult to unroll.

Additionally, and more importantly, many applications require further structural constraints on the learned matrix. Elementwise sparsity is a typical example of such constraints (Banerjee et al., 2008),

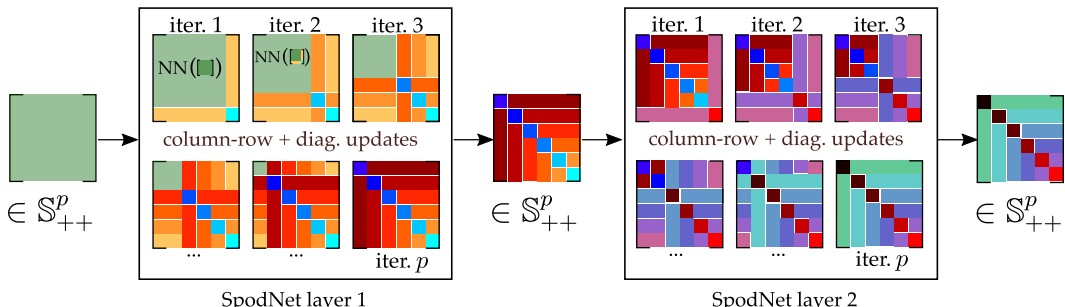

Figure 1: A SpodNet layer chains $p$ updates of column-row pairs and diagonals using neural networks. The matrices remain SPD at all times via Schur's condition.

which significantly increases the complexity of the task: learning functions that simultaneously enforce SPDness and sparsity of the output is known to be challenging (Guillot and Rajaratnam, 2015; Sivalingam, 2015). *There are currently no neural architectures that enable learning jointly SPD and sparse matrices.*

In this paper, we bridge this gap and make the following contributions:

- We introduce a new SPD-to-SPD neural network architecture that also supports enforcing additional constraints on the output. We refer to it as SpodNet for *Schur's Positive-Definite Network*. As a particular case, we show that SpodNet is able to learn jointly SPD and sparse matrices. To the best of our knowledge, **SpodNet is the first architecture to provide strict guarantees for both properties**.

- We demonstrate the framework's relevance through applications in sparse precision matrix estimation. We highlight the limitations of other learning-based approaches and show how SpodNet addresses these issues. Our experiments validate SpodNet's effectiveness in jointly learning SPD-to-SPD and sparsity-inducing functions, yielding competitive results across various performance metrics.

**Notation**   We reserve the bold uppercase for matrices $\boldsymbol{\Theta}$, bold lowercase $\boldsymbol{\theta}$ for vectors and standard lowercase $\theta$ for scalars. The soft-thresholding function is $\mathrm{ST}_\gamma(\cdot) = \mathrm{sign}(\cdot)\max(|\cdot|-\gamma, 0)$ for $\gamma \geq 0$; it acts elementwise on vectors or matrices. On matrices, $\|\cdot\|_1$ is the sum of absolute values of the matrix coefficients. The cone of $p$ by $p$ SPD matrices is denoted $\mathbb{S}_{++}^p$.

## 2   RELATED WORKS

We first review existing approaches for estimating SPD matrices, which can be broadly divided into three categories, all suffering the same limitation of not being able to handle additional structural constraints.

**Riemannian approaches**   Riemannian optimization provides tools to build algorithms whose iterates lie on Riemannian manifolds, such as the SPD manifold (Absil et al., 2008). Many have adopted these tools to design neural architectures that operate on the manifold of SPD matrices (Huang and Van Gool, 2017; Gao et al., 2020; 2022). However, a common and significant bottleneck of these methods is the use of Riemannian operators, which are notoriously expensive to compute. Beyond this computational hindrance, and more importantly, current Riemannian methods are not able to impose additional sparsity on the learned matrices without breaking the SPD guarantee.

**SPD layers**   Neural layers with SPD outputs have also been proposed in Dong et al. (2017); Nguyen et al. (2019). A first line of work rely on the linear mapping $\boldsymbol{X}_{k+1} = \boldsymbol{W}_k \boldsymbol{X}_k \boldsymbol{W}_k^\top$ (with the layer's weights $\boldsymbol{W}_k$ having full row-rank), while some others rely on clipping the eigenvalues of their output. Unfortunately, there are no obvious ways to incorporate additional structure such as elementwise sparsity on the matrices without risking breaking their SPD guarantee. For instance, hard-thresholding the off-diagonal entries of a SPD matrix does not in general preserve the SPD property (Guillot and

Rajaratnam, 2012) and, more generally, elementwise functions preserving this property are very limited (Guillot and Rajaratnam, 2015).

**Unrolled neural architectures** In order to ensure that a network's output strictly respects some desired properties, a successful direction is to unroll convex optimization algorithms (Chen et al., 2022; Shlezinger et al., 2023). This unrolling procedure acts as an "inductive bias" on the architecture, naturally forcing the model to explore suitable solutions within the space of imposed properties. In SPD learning, this approach has been exploited by Shrivastava et al. (2020) who unrolled an optimization algorithm to train neural networks to estimate inverses of covariance matrices. Whilst algorithm unrolling is a very powerful approach to learn specific matrices, it proves difficult to actually enforce several constraints simultaneously on these matrices. We provide further details about the limits of this approach in Section 4.1.

## 3 THE SPODNET FRAMEWORK

We now introduce our core contribution, the SpodNet layer. Essentially, it is an SPD-to-SPD mapping parameterized by neural networks. A SpodNet layer operates by cycling through the $p$ column-row pairs individually by

    (i) updating the corresponding column-row with a neural network,

    (ii) updating the diagonal element to satisfy the SPD constraint (see Figure 1).

Crucially, the neural network used at step *(i)* can update the column-row to *any* value without compromising the SPD guarantee. Consequently, one is free to exploit a spectrum of approaches for those updates, depending on other desired structural properties of the output matrix. In particular, we describe in Section 4 three specific implementations of SpodNet that enforce elementwise sparsity for learning sparse precision matrices.

---

**Algorithm 1** The SpodNet layer

1: Input: $\boldsymbol{\Theta}_{\text{in}} \in \mathbb{S}_{++}^p$ and $\boldsymbol{W}_{\text{in}} = \boldsymbol{\Theta}_{\text{in}}^{-1}$
2: **for** column $i \in \{1, \cdots, p\}$ **do**
3:      Extract blocks: $\boldsymbol{W}_{11}, \boldsymbol{w}_{12}, w_{22}$
4:      Compute $[\boldsymbol{\Theta}_{11}]^{-1} = \boldsymbol{W}_{11} - \frac{1}{w_{22}} \boldsymbol{w}_{12} \boldsymbol{w}_{12}^\top$
5:      New column $\boldsymbol{\theta}_{12}^+ = f(\boldsymbol{\Theta})$
6:      New diagonal value $\theta_{22}^+ = g(\boldsymbol{\Theta}) + \boldsymbol{\theta}_{12}^{+\top} [\boldsymbol{\Theta}_{11}]^{-1} \boldsymbol{\theta}_{12}^+$
7:      Update $\boldsymbol{\Theta} = \boldsymbol{\Theta}^+, \boldsymbol{W} = \boldsymbol{W}^+$ as in Equations (2) and (3)
8: **end for**
9: Output: $\boldsymbol{\Theta}_{\text{out}} \in \mathbb{S}_{++}^p$ and $\boldsymbol{W}_{\text{out}} = \boldsymbol{\Theta}_{\text{out}}^{-1}$

---

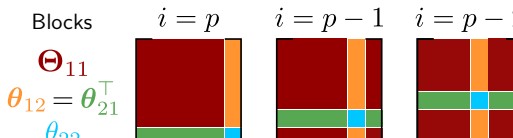

Blocks    $i = p$    $i = p-1$    $i = p-2$

$\boldsymbol{\Theta}_{11}$
$\boldsymbol{\theta}_{12} = \boldsymbol{\theta}_{21}^\top$
$\theta_{22}$

### 3.1 ALGORITHMIC FOUNDATIONS

The key to preserving the positive-definiteness $\boldsymbol{\Theta} \in \mathbb{S}_{++}^p$ of the matrix upon after changing its $i$-th column and row is an appropriate update of the diagonal entry $\Theta_{ii}$ based on Schur's condition for positive-definiteness. In the following, a + superscript on a variable (e.g. $\boldsymbol{\theta}^+$) indicates an update value of this variable (e.g. outputted by a neural network). The following proposition shows that SpodNet's output is guaranteed to be SPD.

**Proposition 3.1.** *Suppose the updated column-row pair is the last one ($i = p$). We partition $\boldsymbol{\Theta}$ as*

$$\boldsymbol{\Theta} = \begin{bmatrix} \boldsymbol{\Theta}_{11} & \boldsymbol{\theta}_{12} \\ \boldsymbol{\theta}_{21} & \theta_{22} \end{bmatrix}, \quad with \quad \boldsymbol{\Theta}_{11} \in \mathbb{R}^{(p-1)\times(p-1)}, \boldsymbol{\theta}_{12} \in \mathbb{R}^{p-1}, \boldsymbol{\theta}_{21} = \boldsymbol{\theta}_{12}^\top, \theta_{22} \in \mathbb{R}. \quad (1)$$

*(for a generic column $i$, $\boldsymbol{\Theta}_{11}$ refers to $\boldsymbol{\Theta}$ without its $i$-th row and $i$-th column, $\boldsymbol{\theta}_{12}$ is the $i$-th row of $\boldsymbol{\Theta}$ without its $i$-th value, and $\theta_{22}$ is $\Theta_{ii}$ as illustrated below Algorithm 1).*

*Suppose that $\boldsymbol{\Theta} \in \mathbb{S}_{++}^p$. Let $\boldsymbol{u} \in \mathbb{R}^{p-1}$ and $v > 0$ be any vector and strictly positive scalar respectively. Then, updating the $i$-th row and column of $\boldsymbol{\Theta}$ as*

$$\boldsymbol{\Theta}^+ \triangleq \begin{bmatrix} \boldsymbol{\Theta}_{11} & \boldsymbol{\theta}_{12}^+ \triangleq \boldsymbol{u} \\ \boldsymbol{\theta}_{21}^+ \triangleq \boldsymbol{u}^\top & \theta_{22}^+ \triangleq v + \boldsymbol{u}^\top [\boldsymbol{\Theta}_{11}]^{-1} \boldsymbol{u} \end{bmatrix}, \quad (2)$$

*preserves the SPD property, i.e. $\boldsymbol{\Theta}^+ \in \mathbb{S}_{++}^p$.*

Since positivity of $\boldsymbol{\Theta}^+$ is guaranteed for any choice of $\boldsymbol{u}$ and $v$ as long as $v > 0$, the principle of SpodNet updates is to *learn* these quantities as function of the current iterate, i.e. from the value of $\boldsymbol{\Theta}$, $\boldsymbol{\Theta}_{11}$, $\boldsymbol{\theta}_{12}$, etc (explicit forms for each variant will be given in Section 4.2). For simplicity, though they can depend on additional information, we write $\boldsymbol{u} = f(\boldsymbol{\Theta})$ and $v = g(\boldsymbol{\Theta})$, with $f$ and $g$ being learned mappings.

*Proof.* Since $\boldsymbol{\Theta}$ is symmetric positive-definite, so is its leading principal submatrix $\boldsymbol{\Theta}_{11}$. It follows that $\boldsymbol{\Theta}^+$ is well-defined, and obviously symmetric. Its positive-definiteness ensues from Schur's condition for positive-definiteness. Indeed $\boldsymbol{\Theta}^+$ is SPD when $\boldsymbol{\Theta}_{11} \succ 0$ and $\theta_{22}^+ - \boldsymbol{\theta}_{12}^{+\top}[\boldsymbol{\Theta}_{11}]^{-1}\boldsymbol{\theta}_{12}^+ > 0$ (Zhang, 2006, Theorem 1.12). The latter condition is ensured as the left hand side is equal to $v$, which is strictly positive by assumption. □

Finally, one SpodNet layer chains $p$ updates of the form Equation (2), sequentially updating all column-row pairs one after the other as summarized in Algorithm 1. A full SpodNet architecture then stacks $K$ SpodNet layers, and, in practice, the first layer takes as input $\boldsymbol{\Theta}_{\text{in}} = (\boldsymbol{S} + \mathbf{I}_p)^{-1}$ where $\boldsymbol{S}$ is the empirical covariance matrix. The overall architecture is schematized in Figure 1 for $K = 2$.

The expressivity of SpodNet comes from the flexibility to use *any* arbitrary functions $f$ and $g$ in the updates whilst guaranteeing that $\boldsymbol{\Theta}$ remains in the SPD cone. Namely, additional structure such as sparsity can trivially be imposed on $\boldsymbol{\Theta}$ through $f$. Broadly speaking, constraints related to the values of the off-diagonal entries of $\boldsymbol{\Theta}$ and that can be imposed on the outputs of a neural network can be controlled directly. We next detail the computational cost of fully updating $\boldsymbol{\Theta}$ through one SpodNet layer.

## 3.2 Improving the update complexity

The diagonal update of Equation (2) a priori requires inverting the matrix $\boldsymbol{\Theta}_{11}$ which comes with a prohibitive cost of $\mathcal{O}(p^3)$ for each column-row update. Fortunately, we are able to leverage the column-row structure of the updates to decrease the cost to a mere $\mathcal{O}(p^2)$, by maintaining the matrix $\boldsymbol{W} = \boldsymbol{\Theta}^{-1}$ up-to-date along the iterations.

**Proposition 3.2.** *Let $\boldsymbol{W}$ be the inverse of $\boldsymbol{\Theta}$, adopting the same block structure $\boldsymbol{W} = \begin{bmatrix} \boldsymbol{W}_{11} & \boldsymbol{w}_{12} \\ \boldsymbol{w}_{12}^\top & w_{22} \end{bmatrix}$. Then $[\boldsymbol{\Theta}_{11}]^{-1} = \boldsymbol{W}_{11} - \frac{1}{w_{22}}\boldsymbol{w}_{12}\boldsymbol{w}_{12}^\top$. In addition, if $\boldsymbol{\Theta}$ is updated as $\boldsymbol{\Theta}^+$ following Equation (2) with $v = g(\boldsymbol{\Theta})$, then the update of $\boldsymbol{W}$ defined by*

$$\boldsymbol{W}^+ \triangleq \begin{bmatrix} [\boldsymbol{\Theta}_{11}]^{-1} + \frac{[\boldsymbol{\Theta}_{11}]^{-1}\boldsymbol{\theta}_{12}^+\boldsymbol{\theta}_{21}^+[\boldsymbol{\Theta}_{11}]^{-1}}{g(\boldsymbol{\Theta})} & -\frac{[\boldsymbol{\Theta}_{11}]^{-1}\boldsymbol{\theta}_{12}^+}{g(\boldsymbol{\Theta})} \\ \left(-\frac{[\boldsymbol{\Theta}_{11}]^{-1}\boldsymbol{\theta}_{12}^+}{g(\boldsymbol{\Theta})}\right)^\top & 1/g(\boldsymbol{\Theta}) \end{bmatrix}. \tag{3}$$

*can be computed in $\mathcal{O}(p^2)$ and satisfies $[\boldsymbol{W}^+]^{-1} = \boldsymbol{\Theta}^+$.*

*Proof.* By the Banachiewicz inversion formula on Schur's complement (Zhang, 2006, Thm 1.2),

$$\begin{bmatrix} \boldsymbol{W}_{11} & \boldsymbol{w}_{12} \\ \boldsymbol{w}_{21} & w_{22} \end{bmatrix} = \begin{bmatrix} \boldsymbol{\Theta}_{11} & \boldsymbol{\theta}_{12} \\ \boldsymbol{\theta}_{21} & \theta_{22} \end{bmatrix}^{-1} = \begin{bmatrix} [\boldsymbol{\Theta}_{11}]^{-1} + \frac{[\boldsymbol{\Theta}_{11}]^{-1}\boldsymbol{\theta}_{12}\boldsymbol{\theta}_{21}[\boldsymbol{\Theta}_{11}]^{-1}}{\theta_{22}-\boldsymbol{\theta}_{21}[\boldsymbol{\Theta}_{11}]^{-1}\boldsymbol{\theta}_{12}} & -\frac{[\boldsymbol{\Theta}_{11}]^{-1}\boldsymbol{\theta}_{12}}{\theta_{22}-\boldsymbol{\theta}_{21}[\boldsymbol{\Theta}_{11}]^{-1}\boldsymbol{\theta}_{12}} \\ \left(-\frac{[\boldsymbol{\Theta}_{11}]^{-1}\boldsymbol{\theta}_{12}}{\theta_{22}-\boldsymbol{\theta}_{21}[\boldsymbol{\Theta}_{11}]^{-1}\boldsymbol{\theta}_{12}}\right)^\top & \frac{1}{\theta_{22}-\boldsymbol{\theta}_{21}[\boldsymbol{\Theta}_{11}]^{-1}\boldsymbol{\theta}_{12}} \end{bmatrix}. \tag{4}$$

Identifying all blocks (namely $g(\Theta)$ with $\theta_{22} - \boldsymbol{\theta}_{21}[\boldsymbol{\Theta}_{11}]^{-1}\boldsymbol{\theta}_{12}$, $\boldsymbol{w}_{12}$ with $-\frac{[\boldsymbol{\Theta}_{11}]^{-1}\boldsymbol{\theta}_{12}}{\theta_{22}-\boldsymbol{\theta}_{21}[\boldsymbol{\Theta}_{11}]^{-1}\boldsymbol{\theta}_{12}}$ and $w_{22}$ with $\frac{1}{g(\boldsymbol{\Theta})}$) yields $[\boldsymbol{\Theta}_{11}]^{-1} = \boldsymbol{W}_{11} - \frac{1}{w_{22}}\boldsymbol{w}_{12}\boldsymbol{w}_{12}^\top$ which can be computed in $\mathcal{O}(p^2)$ if one has access to $\boldsymbol{W}$. This can be achieved using Equation (3), and involves only operations in $\mathcal{O}(p^2)$. The property $\boldsymbol{\Theta}^+ = [\boldsymbol{W}^+]^{-1}$ is satisfied by using the same inversion formula. □

A full update of $\boldsymbol{\Theta}$ can thus be achieved with cost $\mathcal{O}(p^3)$ ($p$ column-row updates of cost $\mathcal{O}(p^2)$).

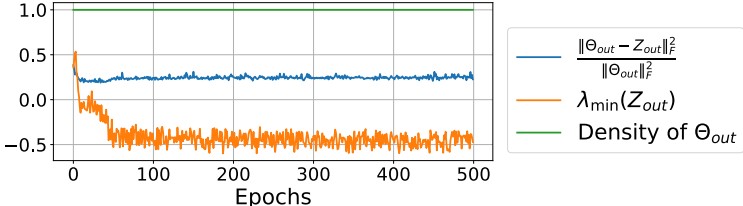

Figure 2: **GLAD's limitations**. Smallest eigenvalue (orange), density degree (green) and relative discrepancy of the two matrices (blue) for an output $(\boldsymbol{Z}_{\text{out}}, \boldsymbol{\Theta}_{\text{out}})$ of GLAD. $\boldsymbol{\Theta}_{\text{out}}$ and $\boldsymbol{Z}_{\text{out}}$ are different, $\boldsymbol{\Theta}_{\text{out}}$ is not sparse, and $\boldsymbol{Z}_{\text{out}}$ is not positive-definite.

## 4 USING SPODNET TO LEARN SPARSE PRECISION MATRICES

So far, we have not specified which choices of column and diagonal updates $f(\boldsymbol{\Theta})$ and $g(\boldsymbol{\Theta})$ we would use in practice. We now leverage the general framework of the SpodNet layer for learning SPD matrices with additional structure and propose three specific architectures for learning **sparse and SPD matrices**, with applications to sparse precision matrix learning.

### 4.1 INFERRING SPARSE PRECISION MATRICES

Consider a dataset of $n$ observed signals $\boldsymbol{x}_1, \ldots, \boldsymbol{x}_n$ where each $\boldsymbol{x}_i \in \mathbb{R}^p$ follows a certain (centered) distribution with covariance matrix $\boldsymbol{\Sigma} \in \mathbb{S}_{++}^p$. The problem of precision matrix estimation arises when attempting to identify the conditional dependency graph of the $p$ variables of this dataset. In Gaussian graphical models (Lauritzen, 1996), this graph is associated with the so-called precision matrix $\boldsymbol{\Theta} = \boldsymbol{\Sigma}^{-1} \in \mathbb{S}_{++}^p$. In this case $\Theta_{ij} = 0$ iff the variables $i$ and $j$ are conditionally independent given the other variables. This estimation problem involves determining $\boldsymbol{\Theta}$ given the empirical covariance matrix $\boldsymbol{S} = \frac{1}{n} \sum_{i=1}^{n} \boldsymbol{x}_i \boldsymbol{x}_i^\top$. In most practical scenarios, the matrix $\boldsymbol{\Theta}$ is sparse due to limited conditional dependencies, which leads to a sparse SPD matrix estimation problem. For this problem, a very popular estimator is the Graphical Lasso (GLasso) (Banerjee et al., 2008; Friedman et al., 2008) that solves

$$\min_{\boldsymbol{\Theta} \succ 0} -\log\det(\boldsymbol{\Theta}) + \langle \boldsymbol{S}, \boldsymbol{\Theta} \rangle + \lambda \|\boldsymbol{\Theta}\|_{1,\text{off}}, \tag{5}$$

where $\|\boldsymbol{\Theta}\|_{1,\text{off}}$ denote the off-diagonal $\ell_1$ norm of $\boldsymbol{\Theta}$, equal to $\sum_{i \neq j} |\Theta_{ij}|$. The data fidelity term $-\log\det(\boldsymbol{\Theta}) + \langle \boldsymbol{S}, \boldsymbol{\Theta} \rangle$ is the negative log-likelihood under Gaussian assumption, while the $\ell_1$ penalty enforces sparsity. It can be efficiently computed (Rolfs et al., 2012; Mazumder and Hastie, 2012; Hsieh et al., 2014) but suffers from the known limitations from the model-based approaches (Adler and Öktem, 2018; Arridge et al., 2019). To circumvent these limitations, two learning-based approaches have been proposed. Belilovsky et al. (2017) introduced DeepGraph, a CNN that directly maps empirical covariance matrices to the graph structures (i.e. to the support of the precision matrix). We emphasize that this model *does not estimate* $\boldsymbol{\Theta}$ but only its support. Shrivastava et al. (2020) proposed GLAD, a neural architecture based on unrolling an alternating minimization algorithm for solving the GLasso. Based on a Lagrangian relaxation, GLAD iterates over two matrices $\boldsymbol{Z}$ and $\boldsymbol{\Theta}$ while learning step-size and thresholding parameters (see Appendix B.2 for more details). Out of these two matrices $\boldsymbol{Z}$ is sparse and $\boldsymbol{\Theta}$ is SPD, but GLAD fails to learn a single matrix that is both sparse and SPD. This limitation is highlighted in Figure 2 showing that, during training, $\boldsymbol{\Theta}$ and $\boldsymbol{Z}$ differ, $\boldsymbol{\Theta}$ is not sparse and $\boldsymbol{Z}$ is not SPD.

### 4.2 SPODNET FOR SPARSE SPD LEARNING

We now show how SpodNet overcomes the limitations of current methods for learning both sparse and SPD matrices. By order of complexity, we introduce three new neural architectures, each corresponding to a specific choice of the functions $f, g$ for learning the values of $\boldsymbol{u}$ and $v$ in Equation (2).

Our choices for $f, g$ are inspired by an unrolling of a proximal block coordinate descent applied to the GLasso problem, where the blocks are column-row pairs described in Section 3. A proximal coordinate gradient descent step on the GLasso objective updates the value of $\boldsymbol{\theta}_{12}$ with

$$\boldsymbol{\theta}_{12}^+ = \text{ST}_{\gamma\lambda}(\boldsymbol{\theta}_{12} - \gamma(\boldsymbol{s}_{12} - \boldsymbol{w}_{12})), \tag{6}$$

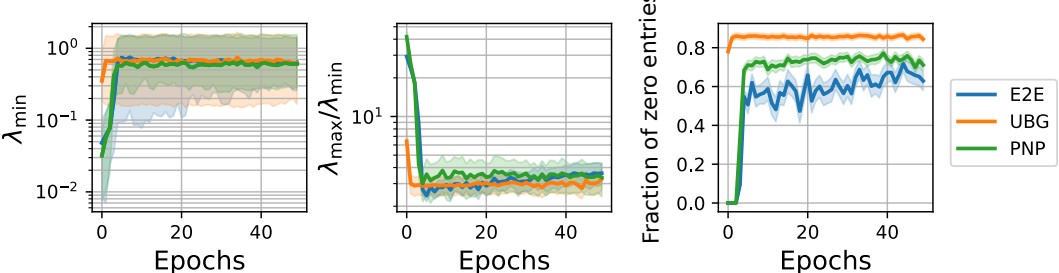

Figure 3: Training dynamics of our 3 models (on test data described in Section 5.1). *Left:* The outputs of our 3 models remain positive-definite. *Middle:* The conditioning remains stable. *Right:* The outputs are sparse. Overall our models produce **jointly sparse + SPD outputs**.

where $\gamma > 0$ is a step-size. This stems from the fact that the gradient of $\boldsymbol{\Theta} \mapsto -\log\det(\boldsymbol{\Theta}) + \langle \boldsymbol{S}, \boldsymbol{\Theta} \rangle$ is $-\boldsymbol{\Theta}^{-1} + \boldsymbol{S}$ (Boyd and Vandenberghe, 2004, Section A.4.1), which restricted to the $\boldsymbol{\theta}_{12}$-block gives $-[\boldsymbol{\Theta}^{-1}]_{12} + \boldsymbol{s}_{12} = -\boldsymbol{w}_{12} + \boldsymbol{s}_{12}$. Together with the fact that the proximal operator of the $\ell_1$ norm is the soft-thresholding (Parikh et al., 2014) we get Equation (6). Because (full) proximal gradient descent on the GLasso is called Graphical ISTA (Rolfs et al., 2012), we coin Equation (6) Block-Graphical ISTA. We can now introduce our three architectures.

**Unrolled Block Graphical-ISTA (UBG)**  First, we propose to learn the stepsizes and soft-thresholding levels when unrolling the Block Graphical ISTA iterations Equation (6). In a learning-based approach, we use as updating function $f$,

$$f_{\text{UBG}} : \boldsymbol{\theta}_{12} \mapsto \text{ST}_{\boldsymbol{\lambda}^+}(\boldsymbol{\theta}_{12} - \gamma^+(\boldsymbol{s}_{12} - \boldsymbol{w}_{12})),\qquad(7)$$

where the step-size $\gamma^+ = \text{NN}_1(\boldsymbol{\theta}_{12}) > 0$ and the soft-thresholding parameters $\boldsymbol{\lambda}^+ = \text{NN}_2(\boldsymbol{\theta}_{12} - \gamma^+(\boldsymbol{s}_{12} - \boldsymbol{w}_{12})) \in \mathbb{R}^{p-1}$. $\text{NN}_1$ and $\text{NN}_2$ are small multilayer perceptrons with architectures provided in Appendix A.2. We emphasize that this is different from hyperparameter tuning, since $\gamma^+$ and $\boldsymbol{\lambda}^+$ are predicted at each update instead of being global parameters. UBG exhibits the highest inductive bias among the models, as its architecture is directly derived from unrolling an iterative optimization algorithm designed to minimize a model-based loss.

As highlighted in Section 3, one can plug *any* update functions and still get SPD outputs. This motivates extending UBG to a more expressive architecture.

**Plug-and-Play Block Graphical-ISTA (PNP)**  Precisely, we extend UBG to a Plug-and-Play-like setting (Venkatakrishnan et al., 2013; Romano et al., 2017). In a nutshell, these methods replace the proximal operator in first-order algorithms by a denoiser, usually implemented by a neural network. In our context, this corresponds to replacing the soft-thresholding of UBG by a neural network $\Psi : \mathbb{R}^{p-1} \to \mathbb{R}^{p-1}$, that is,

$$f_{\text{PnP}} : \boldsymbol{\theta}_{12} \mapsto \Psi(\boldsymbol{\theta}_{12} - \gamma^+(\boldsymbol{s}_{12} - \boldsymbol{w}_{12})).\qquad(8)$$

To promote sparsity of the output of $f_{\text{PnP}}$, the last layer of $\Psi$ performs an elementwise soft-thresholding. The parameter for this soft-thresholding is also learned from data and given by the same multilayer perceptron that predicts $\boldsymbol{\lambda}^+$ in UBG (Appendix A.2).

**End-to-end updates (E2E)**  Finally, we propose a fully-flexible architecture without any algorithm-inspired assumptions. Precisely, we consider

$$f_{\text{E2E}} : \boldsymbol{\theta}_{12} \mapsto \Phi(\boldsymbol{\theta}_{12}),\qquad(9)$$

where $\Phi$ takes in the current state of the column $\boldsymbol{\theta}_{12}$ and learns to predict an adequate column update $\boldsymbol{\theta}_{12}^+$. Intuitively, the neural network $\Phi$ acts as learning both the forward and the backward steps of a forward-backward iteration (Combettes and Pesquet, 2011). As for PNP, sparsity of the predictions is enforced by a soft-thresholding non-linearity in the last layer of $\Phi$.

Finally, for all models, we use for $g$ a small neural network that takes as input $\theta_{22}, s_{22}$ and $(\boldsymbol{\theta}_{12}^+)^\top[\boldsymbol{\Theta}_{11}]^{-1}\boldsymbol{\theta}_{12}^+$. Its positivity is ensured by using an absolute value function as final nonlinearity. These input features are based on the intuition that the network should exploit information

from the current state of the diagonal of $\boldsymbol{\Theta}$, the diagonal of $\boldsymbol{S}$ which is closely related to the diagonal of the GLasso estimator at convergence, and Schur's complement which is added to the output of $g$.

**Improving training stability**    Although the positive definiteness of $\boldsymbol{\Theta}$ is guaranteed to be preserved at each update for any positive-valued function $g$, we have empirically observed that its smallest eigenvalue could approach 0 as visible in Appendix B.1. In practice this leads to instability in training. Below, we provide an interpretation of this phenomenon and provide a solution to address it. Each column-row update inside a SpodNet layer can be written as the rank-2 update

$$\boldsymbol{\Theta}^+ = \underbrace{\begin{bmatrix} \boldsymbol{\Theta}_{11} & \boldsymbol{\theta}_{12} \\ \boldsymbol{\theta}_{21} & \theta_{22} \end{bmatrix}}_{=\boldsymbol{\Theta}} + \underbrace{\begin{bmatrix} 0 & \boldsymbol{\theta}_{12}^+ - \boldsymbol{\theta}_{12} \\ (\boldsymbol{\theta}_{12}^+ - \boldsymbol{\theta}_{12})^\top & \theta_{22}^+ - \theta_{22} \end{bmatrix}}_{\triangleq \Delta_{\boldsymbol{\Theta}}}. \tag{10}$$

Moreover, the nonzero eigenvalues of the perturbation $\Delta_{\boldsymbol{\Theta}}$ are given by $\lambda_{\pm} = \frac{(\theta_{22}^+ - \theta_{22}) \pm \sqrt{(\theta_{22}^+ - \theta_{22})^2 + 4\|\boldsymbol{\theta}_{12}^+ - \boldsymbol{\theta}_{12}\|^2}}{2}$, and by the Bauer-Fike theorem, we can quantify the evolution of $\boldsymbol{\Theta}$'s spectrum by

$$|\lambda_k(\boldsymbol{\Theta}) - \lambda_k(\boldsymbol{\Theta}^+)| \leq \|\Delta_{\boldsymbol{\Theta}}\|_{\mathrm{op}}, \tag{11}$$

where $\lambda_k$ denotes the $k$-th largest eigenvalue of a SPD matrix. Hence, one way to ensure that the perturbation of the spectrum is small is to control $\|\Delta_{\boldsymbol{\Theta}}\|_{\mathrm{op}}$, for instance, by limiting the magnitude of the updates. Experimentally, the most successful approach to control $\|\Delta_{\boldsymbol{\Theta}}\|_{\mathrm{op}}$ is to limit the magnitude of $\theta_{22}^+$, which is not a trivial task as the updated value $\theta_{22}^+ = g(\boldsymbol{\Theta}) + \boldsymbol{\theta}_{21}^+[\boldsymbol{\Theta}_{11}]^{-1}\boldsymbol{\theta}_{12}^+$ must satisfy $\theta_{22}^+ - \boldsymbol{\theta}_{21}^+[\boldsymbol{\Theta}_{11}]^{-1}\boldsymbol{\theta}_{12}^+ > 0$. Thus, we propose to scale $\boldsymbol{\theta}_{12}^+$ by scaling the preactivation[1] $\boldsymbol{z}$ of the last layer of $f$ by $\frac{\sqrt{\zeta}}{\sqrt{\boldsymbol{z}^\top [\boldsymbol{\Theta}_{11}]^{-1} \boldsymbol{z}}}$. The hyperparameter $\zeta > 0$ acts as a form of regularization that handles a compromise between stability and expressivity of the model; in all of our experiments we use $\zeta = 1$. As visible in Figure 3 this scaling ensures a smooth training of the network, with stable condition number of the $\boldsymbol{\Theta}$ matrix.

## 5 EXPERIMENTS

We now illustrate SpodNet's ability to learn SPD matrices with additional structure by using our three derived graph learning models (UBG, PNP and E2E) to learn sparse precision matrices, on both synthetic and real data. Additional details regarding the experimental setups can be found in Appendix A. `PyTorch` implementations can be found in our GitHub repository[2].

### 5.1 SPARSE PRECISION MATRIX RECOVERY

Our goal in this section is to evaluate our models' performance and generalization ability regarding the recovery of sparse precision matrices on synthetic data. The following experiments serve several purposes: (1) compare learning-based methods against traditional model-based methods, (2) evaluate our models' reconstruction performance in terms of matrix estimation and support recovery.

**Data & training**    We use synthetic data to train our SpodNet models as in Belilovsky et al. (2017); Shrivastava et al. (2020). We generate $N$ sparse SPD $p \times p$ matrices using `sklearn`'s `make_sparse_spd_matrix` function (Pedregosa et al., 2011), of which we ensure proper conditioning by adding $0.1 \cdot \mathbf{I}_p$. These matrices are treated as ground truth precision matrices $\boldsymbol{\Theta}_{\mathrm{true}}^{(i)}$ for $i \in [N]$. The sparsity degree of each matrix is controlled through the one imposed on their Cholesky factors during their generation: in the *strongly sparse* setting $\boldsymbol{\Theta}_{\mathrm{true}}$ has roughly 90 % of zero entries while in the *weakly sparse* setting this value is around 25 %. For each of these $N$ ground truth precision matrices $\boldsymbol{\Theta}_{\mathrm{true}}$, we sample $n$ i.i.d. centered Gaussian random vectors $\boldsymbol{x}_j \sim \mathcal{N}(0, (\boldsymbol{\Theta}_{\mathrm{true}}^{(i)})^{-1})$, which are used to compute an empirical covariance matrix $\boldsymbol{S}^{(i)}$. The dataset thus comprises couples $(\boldsymbol{S}^{(i)}, \boldsymbol{\Theta}_{\mathrm{true}}^{(i)})$ where each $\boldsymbol{S}^{(i)}$ stems from a *different* ground truth precision matrix. For all the experiments, we generate $N_{\mathrm{train}} = 1000$ different $(\boldsymbol{S}, \boldsymbol{\Theta}_{\mathrm{true}})$ couples for the

---

[1]Experimentally scaling the preactivation works better than scaling the network's output.
[2]https://github.com/Perceptronium/SpodNet

training set and $N_{\text{test}} = 100$ couples for the testing set on which we validate our models. Further details on the parameters used during the data generation are in Appendix A.1.

We train our three models (UBG, PNP and E2E) to minimize a reconstruction error $\mathcal{L}_{\text{MSE}} = \frac{1}{N_{\text{train}}} \sum_{i=1}^{N_{\text{train}}} \|\hat{\boldsymbol{\Theta}}^{(i)} - \boldsymbol{\Theta}_{\text{true}}^{(i)}\|_F^2$ in the vein of Gregor and LeCun (2010); Shrivastava et al. (2020), where $\hat{\boldsymbol{\Theta}}$ is the model's output. All three models are trained using ADAM with default hyperparameters (Kingma and Ba, 2014).

**Baselines**   We compare our three models to various other approaches throughout our experiments: learning-based, Riemannian and model-based estimators. As performed in Belilovsky et al. (2017); Shrivastava et al. (2020), we use the three most popular methods for estimating precision matrices as our core baselines: the GLasso (Friedman et al., 2008), the inverse Ledoit-Wolf estimator (Ledoit and Wolf, 2004) and the inverse OAS (Chen et al., 2010). The Ledoit-Wolf and OAS estimators are covariance matrix estimators, for which we take the inverse for estimating the precision (we simply refer to them as Ledoit-Wolf and OAS). All three approaches guarantee the positive-definiteness of the estimated matrix but the GLasso is the only one to additionally ensure sparse predictions. For the GLasso, for each matrix $\boldsymbol{S}^{(i)}$ a sparsity-regulating hyperparameter $\lambda^{(i)}$ is chosen using `sklearn`'s `GraphicalLassoCV` cross-validation procedure, that relies only on the samples used in $\boldsymbol{S}^{(i)}$. In the same vein, we use dedicated Ledoit-Wolf and OAS estimators for each $\boldsymbol{S}^{(i)}$. We also compare our approaches to the GLAD model (Shrivastava et al., 2020), considering both its $\boldsymbol{\Theta}$ and $\boldsymbol{Z}$ outputs. We recall that $\boldsymbol{Z}$ is not guaranteed to be positive-definite and its validity as a precision matrix estimator is therefore debatable. GLAD is trained with the same loss, data and optimizer as our models.

We compare the performance of the different approaches in terms of Normalized MSE on the test set: $\text{NMSE} = \frac{1}{N_{\text{test}}} \sum_{i=1}^{N_{\text{test}}} \frac{\|\hat{\boldsymbol{\Theta}}^{(i)} - \boldsymbol{\Theta}_{\text{true}}^{(i)}\|_F^2}{\|\boldsymbol{\Theta}_{\text{true}}^{(i)}\|_F^2}$ and F1 score for assessing support recovery. For the latter, false positives correspond to $\hat{\boldsymbol{\Theta}}_{ij} \neq 0$, $(\boldsymbol{\Theta}_{\text{true}})_{ij} = 0$ (and similarly for true positives and false negatives).

As shown in Figure 4, at convergence, the learning-based methods show to be more suited for the actual reconstruction of $\boldsymbol{\Theta}_{\text{true}}$ as they significantly outperform traditional methods. This superior performance is especially pronounced as the dimensionality of the data grows and the sparsity of the ground truth decreases. In the most challenging scenario ($p = 100$ and $n = 100$ with weakly sparse $\boldsymbol{\Theta}_{\text{true}}$), the traditional methods perform poorly, with NMSE around 90%.

Figure 4 also suggests that our models perform especially well in terms of NMSE in the low-samples regime ($n \leq p$), which is a common setting in various real-world applications, such as neuroscience time-series analysis or financial temporal network inference. In Figure 5, we thus study the influence of the number of samples $n$ on the performance of each method. The results show that although all methods achieve excellent recovery in the large sample regime (down to around 2.5% NMSE in the lower dimensional $p = 20$ setting), our models are especially more suited for matrix reconstruction in case of sample-deficiency, especially so in the higher dimensional ($p = 100$) setting. In terms of support recovery, our models achieve significantly better F1 scores than the GLasso as the number of sample $n$ grows, with up to 25% improvement in certain tested settings. In the lower dimensional setting ($p = 20$), we are even able to achieve a F1 score of over 0.75 with our UBG model. Finally, although GLAD's $\boldsymbol{Z}$ performs well, an additional experiment in Appendix B.3 shows that those matrices turn out to never be SPD in various settings, making them unsuited as precision matrix estimators.

## 5.2 APPLICATION TO UNSUPERVISED GRAPH LEARNING ON A REAL-WORLD DATASET

We finally evaluate the performance of SpodNet in a graph learning context using a real-world dataset and under an *unsupervised scenario*. The aim of this experiment is twofold: (1) to assess the generalization capabilities of SpodNet, and (2) to determine if it can yield an accurate graph topology *when trained solely on synthetic data*, following the idea of Belilovsky et al. (2017).

**Data & training**   We consider the Animals dataset (Lake and Tenenbaum, 2010), which comprises $p = 33$ animal species, each characterized by responses to $n = 102$ binary questions (e.g., "Has teeth?", "Is poisonous?"). The objective is to infer a graph of connections between these $p$ animals

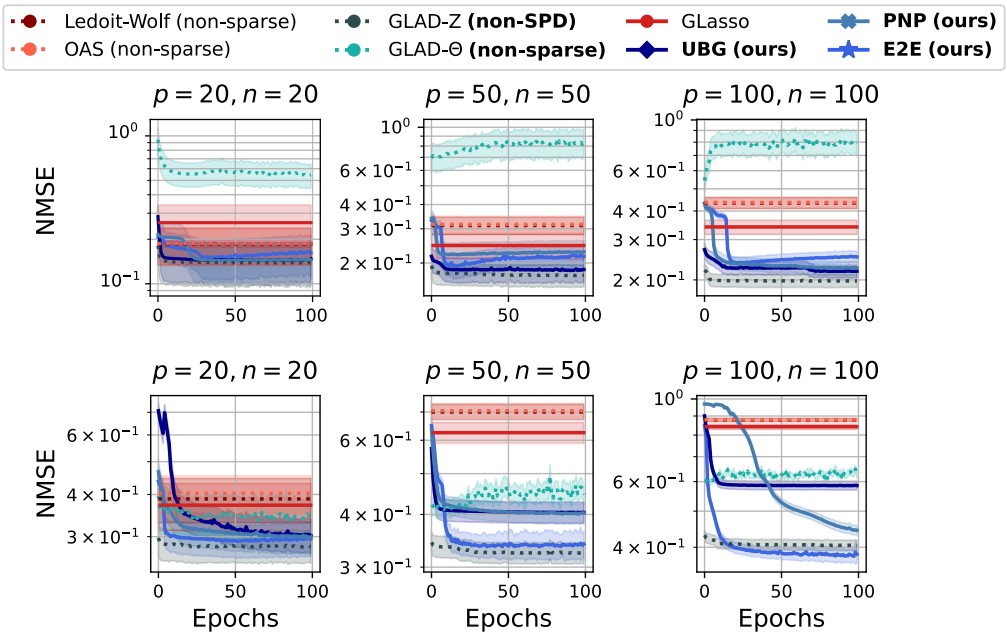

Figure 4: Learning-based (in variations of blue) vs traditional methods (in variations of red). Dotted curves indicate when one of the constraints (SPDness or sparsity) is not guaranteed. *First row:* Strongly sparse $\Theta_{\text{true}}$. *Second row:* Weakly sparse $\Theta_{\text{true}}$.

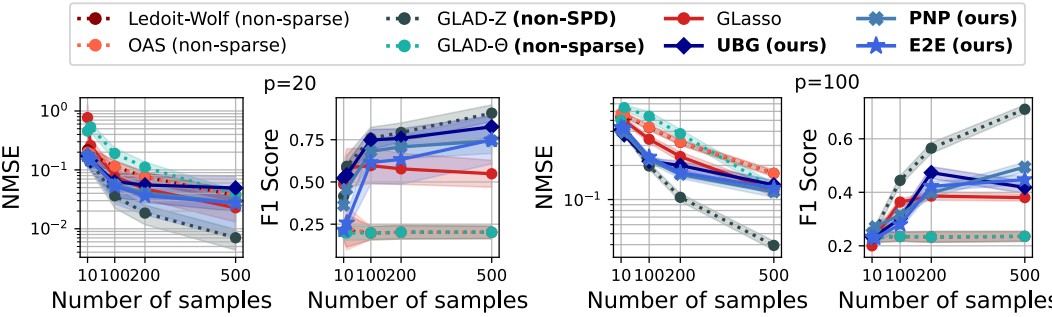

Figure 5: Comparing models in sample deficient regimes ($n \leq p$) up to large-sample regimes ($n \gg p$), evaluated in terms of NMSE and F1 score for support recovery. Dotted curves indicate when one of the constraints (SPDness or sparsity) is not guaranteed. In large dimension, GLAD's $Z$ performs well, but is never SPD; GLAD's $\Theta$ is SPD, but performs badly.

based on the $n$ responses. We train our UBG model by minimizing the MSE on the synthetic data detailed in Section 4.2, with $p = 33, n = 102$ and various levels of sparsity. We generate 1000 training matrices and train the model for 100 epochs. The predicted precision matrix is then computed by inputting the empirical covariance matrix from the Animals dataset into SpodNet.

**Baselines** We compare our approach with the GLasso (with the regularization parameter selected by cross-validation), GLAD (which is trained with the same data as our model), and the Elliptical Graphical Factor Model (EGFM) proposed by Hippert-Ferrer et al. (2023). The latter is a Riemannian optimization method specifically designed to identify clusters within the data by estimating a precision matrix whose inverse is modeled as a low-rank matrix plus a positive diagonal. For a fair comparison, we adopt the same hyperparameters as outlined in the original paper, setting the rank and regularization parameter to 10.

Each method yields a precision matrix, which is used to construct a graph representing the connections between the $p$ animals. The graph is constructed by considering the absolute values of the precision

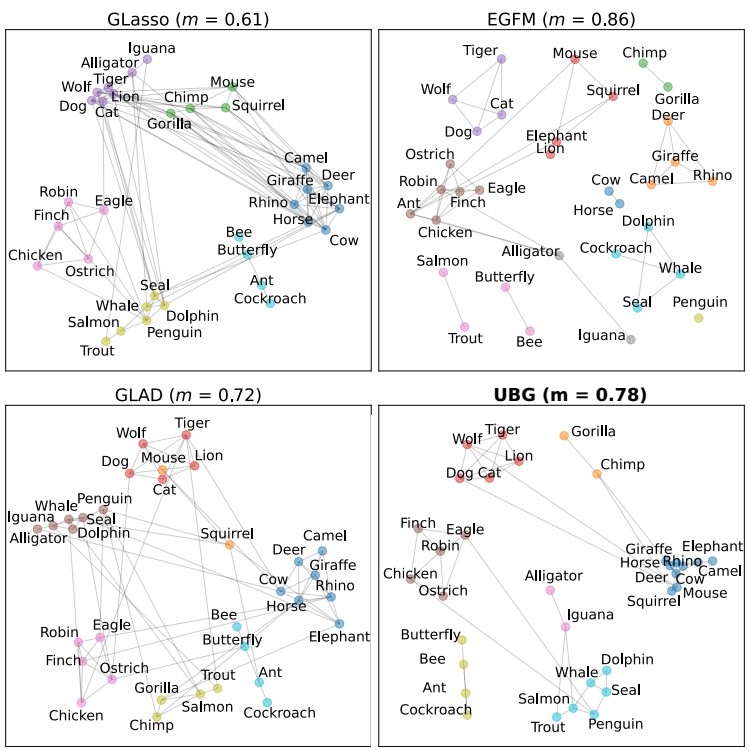

Figure 6: Graph estimation on the Animals dataset with the GLasso, EGFM, GLAD and our UBG model. The last three yield cleaner and more interpretable clusters than the GLasso, but EGFM is specifically tailored for finding clusters as opposed to our UBG model or GLAD.

matrix coefficients, which represent the strength of the connections, and by removing the diagonal elements to eliminate self-loops. The results are illustrated in Figure 6, where the nodes of the graphs are partitioned using the Louvain algorithm (Blondel et al., 2008). Qualitatively, we observe that UBG produces a coherent graph, with outcomes comparable to those of the three baselines. Additionally, the graph learned by our UBG model exhibits a cleaner structure compared to GLasso, with a sparser representation and finer clusters, such as the grouping of (*Chimp*, *Gorilla*). To quantitatively assess the quality of the obtained graphs, we calculate the modularity $m$ of the partitions: higher modularity values indicate better separation of the graph into distinct subcomponents (Newman, 2006). We find that the graph produced by UBG achieves a higher modularity ($m = 0.78$) than GLasso ($m = 0.61$) and GLAD ($m = 0.72$), and slightly lower than EGFM ($m = 0.86$). It is important to note that the EGFM is specifically designed to obtain clustered graphs, whereas our UBG model has no such inductive bias. These combined quantitative and qualitative results demonstrate that SpodNet generalizes effectively to unseen data and produces a coherent graph structure.

## 6 CONCLUSION

We have proposed *Schur's Positive-Definite Network* (SpodNet), a novel learning module compatible with other standard architectures, offering strict guarantees of SPD outputs. The principal novelty of SpodNet comes from its ability to handle additional desirable structural constraints, such as elementwise sparsity which we used as an illutrative example throughout this paper. To the best of our knowledge, SpodNet layers are the first to offer strict guarantees of such highly non-trivial structure in the outputs. In future works, SpodNet could be leveraged to learn other additional structures beyond sparsity. We have shown how to leverage SpodNet to build neural architectures that outperform traditional methods in the context of sparse precision matrix estimation in various settings. Future research will be dedicated to improving the computational cost of our framework, theoretical understanding of the eigenvalues' dynamics during training and of SpodNet's expressivity, and deriving formal convergence guarantees.

## 7    REPRODUCIBILITY STATEMENT

We believe this paper fully discloses the information needed to reproduce the main experimental results. Our core contribution is the SpodNet layer: its general algorithmic framework is presented in Algorithm 1. Three specific models built using this framework for learning sparse and SPD matrices are introduced in Section 4.2, and the exact architectures used for each of them throughout our experiments are explained in detail in Appendix A.2. The conducted experiments are described in Section 5 with additional details in Appendix A. `PyTorch` implementations can be found in our GitHub repository.

## 8    ACKNOWLEDGEMENTS

We would like to thank Badr Moufad (Ecole Polytechnique, France) for his help with the implementations, Paulo Gonçalves (Inria Lyon, France) for fruitful discussions as well as the Centre Blaise Pascal for computing ressources, which uses the SIDUS solution developed by Emmanuel Quemener (ENS Lyon, France) (Quemener and Corvellec, 2013). This work was partially supported by the AllegroAssai ANR-19-CHIA-0009 project.

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

# A    EXPERIMENTAL DETAILS

## A.1    SYNTHETIC DATA GENERATION

Here we provide additional details on the experiment of Section 5.1. The sparsity degree of each matrix is controlled through the one imposed on their Cholesky factors during their generation, with the `alpha` parameter of the function. Throughout our experiments, we consider `alpha` $\in \{0.7, 0.95\}$; this is the probability of a 0 entry on the Cholesky factor of the generated matrix and not the actual fraction of null elements of $\Theta$. Although the numbers vary with the dimension $p$, $\alpha = 0.95$ roughly leads to 90 % of zero entries, while $\alpha = 0.7$ leads to 25 % of zero entries.

## A.2    ARCHITECTURES FOR UBG, PNP AND E2E

The function $g$ for all three models is a MLP with two hidden layers of 3 neurons each with a ReLU activation function. It takes as input $\theta_{22}, s_{22}$ and $(\boldsymbol{\theta}_{12}^+)^\top [\boldsymbol{\Theta}_{11}]^{-1} \boldsymbol{\theta}_{12}^+$.

**UBG**    Its $\gamma^+$ parameter is predicted by a MLP that takes as input the current state of the $\boldsymbol{\theta}_{12}$ block (of dimension $p - 1$) and has a single hidden layer of $\lfloor p/2 \rfloor$ neurons with a ReLU activation function. The (single) output neuron has an absolute value activation, in order to keep it positive since it predicts a step-size.

UBG involves another MLP to learn the vector $\boldsymbol{\lambda}^+$ of elementwise soft-thresholding parameters. This MLP has a single hidden layer of 5 neurons, with ReLU activation function. For the same reason as before, the output layer also has an absolute value activation.

**PNP**    First, the step-size $\gamma^+$ is predicted by a MLP with the exact same architecture as the one in UBG. The learned operator $\Psi$ takes as input $\boldsymbol{\theta}_{12} - \gamma^+(\boldsymbol{s}_{12} - \boldsymbol{w}_{12})$, passes it through a single hidden layer of $2p$ neurons, followed by a ReLU activation function, and projects it back into a $p - 1$ dimensional vector.

The architecture to predict $\boldsymbol{\lambda}$ is the same as for UBG. Our experiments show that multiplying its output by 0.1 helps avoid local minimas and improves convergence.

**E2E**    The neural network $\Phi$ is a MLP that takes as input $\boldsymbol{\theta}_{12}$, passes it through a single hidden layer of $10p$ neurons follow by a ReLU activation function, and projects it back into a $p - 1$ dimensional vector. The architecture to predict $\boldsymbol{\lambda}$ is the same as for UBG and PNP. Our experiments show that multiplying its output by 0.1 helps avoiding local minimas and improves convergence.

**GLAD**    We use the authors' original implementation retrieved from the authors repository at https://github.com/Harshs27/GLAD, with the default hyperparameters.

All four models are implemented with $K = 1$ layer throughout all the experiments, since our results show that this was enough to yield competitive to outperforming results in the settings under consideration.

## A.3    DETAILS ON FIGURES 4 AND 5

For the weakly sparse settings, we use a learning rate of $10^{-3}$ for all three of our own models. For the strongly sparse settings, we use a learning rate of $10^{-2}$. GLAD's learning rate is set to $10^{-2}$ in all settings. All four models are trained on the same 1000 training matrices, using a batch-size of 10, with ADAM's default parameters. The models are tested on the same 100 matrices as mentioned in Section 5.1.

For Figure 5 We use the strongly sparse ground truths in order for the F1 score to be more sensical, and to better illustrate the relevancy of the learned sparsity patterns. We use a learning rate of $10^{-2}$ for all three of them in every setting for $p = 100$, and of $5 \cdot 10^{-3}, 10^{-2}, 10^{-3}, 3 \cdot 10^{-2}$ in the $p = 20$ setting for respectively $n = 10, n = 20, n = 100, n = 200$ and $n = 500$.

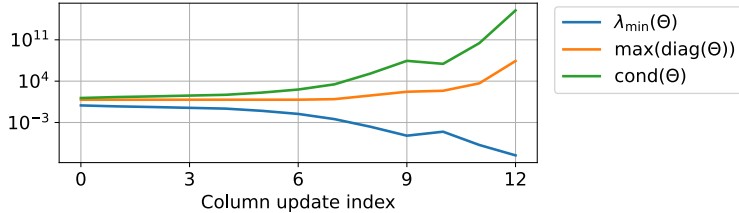

Figure 7: Potential instabilities without normalization. Along column update indices, we plot the smallest eigenvalue (blue), largest diagonal value (orange) and conditioning (green) of a $\Theta$.

## B ADDITIONAL DETAILS ON MODELS

### B.1 SpodNet's potential instabilities without normalization

We show in Figure 7 the evolution of the conditioning of an example of $\Theta$ during training that becomes unstable if we do not use the stabilization procedure described in Section 4. We observe that although the matrix remains SPD as predicted by Proposition 3.1 but gets increasingly closer to being singular during the updates.

### B.2 GLAD's update rules

GLAD unrolls an algorithm that solves the following optimization problem:

$$\min_{\Theta, Z \in \mathbb{S}^p_{++}} - \log \det (\Theta) + \langle S, \Theta \rangle + \lambda \|Z\|_1 + \frac{\alpha}{2} \|Z - \Theta\|_F^2 \,. \tag{12}$$

Each iteration of GLAD, which can be seen as an individual layer in a deep learning perspective, updates several running variables:

$$\alpha^+ = \tilde{f}(\|Z - \Theta\|_F^2, \alpha) \,, \tag{13}$$

$$Y^+ = \frac{1}{\alpha^+} S - Z \,, \tag{14}$$

$$\Theta^+ = \frac{1}{2} \left( -Y^+ + \sqrt{Y^{+\top} Y^+ + \frac{4}{\alpha^+} \mathbf{Id}} \right) \,, \tag{15}$$

$$\lambda_{ij}^+ = \tilde{h}(\Theta_{ij}^+, S_{ij}, Z_{ij}) \,, \tag{16}$$

$$Z_{ij}^+ = \mathrm{ST}_{\lambda_{ij}^+} \left( \Theta_{ij}^+ \right) \,, \ \forall i, j \in [p] \,, \tag{17}$$

in which the functions $\tilde{f} : \mathbb{R}^2 \to \mathbb{R}$ and $\tilde{h} : \mathbb{R}^3 \to \mathbb{R}$ are two small neural networks that are trained to predict adequate parameters for each layer. By construction, $\Theta^+$ is always SPD and a sparsity structure is enforced on $Z^+$, but the converse is not true.

### B.3 GLAD's non-SPD outputs

As Figure 8 shows, in the setting of Figure 4, GLAD's best performing outputs $Z$ are SPD in virtually $0\%$ if the case strongly sparse case with $p = 100$.

Additionally, $Z$ loses its sparsity if projected onto the SPD cone and becomes singular (since it is actually projected onto the closed cone of semi-definite matrices). Going the other way around and trying to sparsify $\Theta$ instead, by thresholding its off-diagonal entries following the support found by $Z$, breaks its positive-definiteness guarantee (Figure 9).

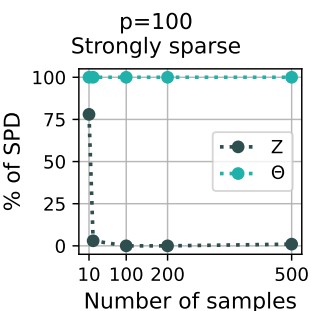

Figure 8: Percentage of SPD outputs among GLAD's $Z$ and $\Theta$ outputs at convergence. As detailed in Section 4.1, $Z$ is not guaranteed to be sparse.

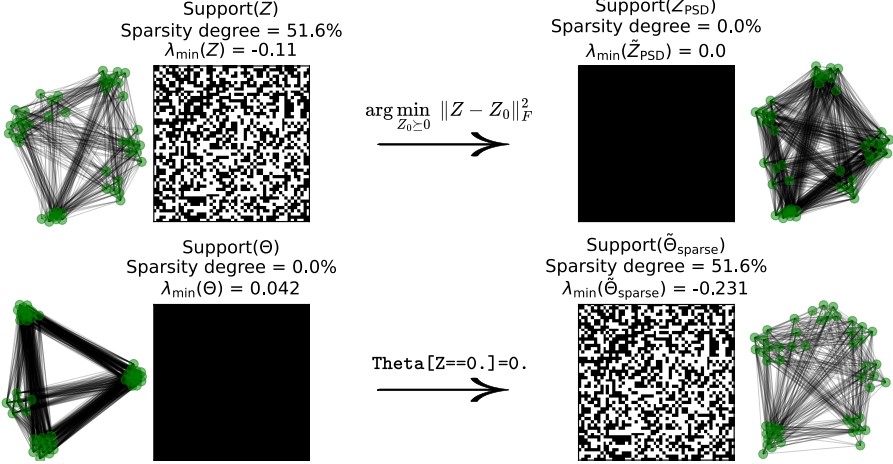

Figure 9: *First row:* Projecting GLAD-$Z$ onto the PSD cone. *Second row:* Thresholding GLAD-$\Theta$ following the support of GLAD-$Z$.

