# OpenReview forum: "Schur's Positive-Definite Network: Deep Learning in the SPD cone with structure"
_ICLR.cc/2025/Conference — ICLR 2025 Poster_

### Official Review · Reviewer_R2Se · 2024-10-29

**Soundness:** 3
**Presentation:** 3
**Contribution:** 3
**Rating:** 6
**Confidence:** 4

**Summary:**

The paper presents a method for jointly learning Symmetric Positive Definite (SPD) and sparse matrices using a learning based approach. The proposed method SpodNet is an iterative method which learns column-row pairs stepwise where each step is parametrized by a neural network. The SPD constraint is then ensured via Schur’s condition.

**Strengths:**

1. SpodNet enables increased expressivity for estimating SPD matrices with a neural network based parametrization, at the same time maintaining constraints like sparsity unlike other model based approaches, as verified by experiments on synthetic data.
2. In order to make the method tractable the authors leverage the block structure of SPD matrices to restrict the complexity of each update step to $O(p^2)$.
3. Experiments on synthetic data and graph topology estimation using SpodNet are presented, highlighting the effectiveness of the method. On the synthetic data, the proposed method is similar to model based approaches while achieving both SPD and sparsity at the same time.

**Weaknesses:**

1. The proposed SpodNet provides a novel approach to leverage neural networks and increase expressivity on constrained manifolds. However, the design of the neural networks itself is not discussed in detail. It is not clear to me why the authors choose the input features for  $g$ as $\theta_{22}$, $s_{22}$ and $\theta_{12} \theta_{11}^{-1} \theta_{12}$. Similar for each of the three described approaches the explanation for choosing the input features is missing. I assume the features are chosen to best suit the model based approach and that performs well with gradient descent but the paper would benefit from a detailed explanation of the same.
2. The method still seems relatively expensive in spite of the improved update rule. The overall cost as the authors mentioned is of the order of $O(p^3)$, how does this compare with the other model based approaches?
3. Can the SpodNet framework maintain other structure constraints for example structural sparsity. In general what conditions would the constraints need to satisfy in order to be optimized with a SpodNet layer.

Since the general literature of SPD matrix estimation points towards applications in computer vision, it would be informative to see an experiment for a vision task with SpodNet to verify the comparison with baselines and its scalability given its computational requirements.

**Questions:**

See above

---

> ### Author Response · Authors · 2024-11-20
> **Response to R2Se**
>
> Dear reviewer,
>
> Thank you for your comments and interesting questions, we are happy that you enjoyed reading the paper. Here is our response:
>
> 1. **Your intuition that the input features are chosen to best suit the model-based approach and to perform well with GD is correct, we have added more explanations in Section 4.2 (lines 311-314).** Regarding the input features of $f$, their core motivation is to impose an inductive bias on models (as addressed in lines 269, 284-286) in the case of UBG and PNP. In the case of E2E, it was drawn from the intuition that the neural network should freely exploit the current state of the column in order to produce an appropriate update (lines 302-305). The input features for $g$ are design choices that were made by inspiration from the $\lambda$ estimator of GLAD [1], and that more importantly showed the best empirical performance. It is based on the intuition that the network should exploit information from the current state of the diagonal of $\Theta$, the diagonal of $S$ which is closely related to the diagonal of the GLasso estimator at convergence, and Schur’s complement which is added to the output of $g$.
> 2. The overall cost is of the order of $\mathcal{O}(p^3)$ complexity for a complete update, as you state. This cost is the same as an update of $\Theta$ for many other model based solvers, e.g. Graphical-ISTA or Graphical Lasso, and makes inference quite fast (less than 1 s, which is similar to solving one GLasso). As always with learning-based methods, the cost of training on many examples is more important, in the order of hours ; future directions involve making training faster.
> 3. Yes, structural sparsity could be enforced through $f$. **Broadly speaking, any constraint related to the values of the off-diagonal entries, and that can be imposed on the outputs of a neural network, can be imposed with $f$**.  In the case of sparsity with prior known structure, where some predefined elements are strictly required to be non-zero, a straight-forward approach would be to discard the sparsity-enforcing operation on the appropriate entries. Alternatively, if the desired structure imposes predefined off-diagonal elements to be strictly equal to 0, these entries can be hard-coded to be so. Other non-trivial constraints such as structured sparsity with unknown prior structure could be tackled with neural networks that are designed to produce structured sparse outputs (e.g. in [2]).
> 4. We agree that this would be a very interesting application of SpodNet and definitely consider this as a potential future direction of research ! However, due to time constraints, it is difficult to design and implement relevant experiments in computer vision in this rebuttal period.
>
> _Refs:_
>
> [1]: GLAD: Learning Sparse Graph Recovery. Shrivastava et al., 2020.
>
> [2]: Feature selection using a neural network with group lasso regularization. Zhang et al., 2020.

---

> > ### Comment · Reviewer_R2Se · 2024-11-21
> > **Response to author rebuttal**
> >
> > I thank the authors for providing clarifications for all my concerns.
> >
> > 1. The additional explanation in lines 311-314 does help improve the overall understanding.
> >
> > 3. Maybe this can be added in the paper regarding additional kinds of constraints which can be optimized with SpodNet?
> >
> > I am happy with the authors' response and don't have further questions. I will keep my score.

---

> > > ### Author Response · Authors · 2024-11-27
> > >
> > > Dear reviewer,
> > >
> > > We are glad to read this sentiment. We have updated the paper according to your suggestion (lines 169-170). We thank you for your valuable input !

---

### Official Review · Reviewer_MbyP · 2024-11-02

**Soundness:** 2
**Presentation:** 3
**Contribution:** 2
**Rating:** 6
**Confidence:** 4

**Summary:**

This paper proposes a deep learning-based approach to solving SPD problems such as covariance selection. The authors start with the block coordinate descent (BCD) algorithm and then unroll the optimization using neural networks. This follows with three different unrollings, where each preserves different levels of the problem structures (or inductive bias in the deep learning words). The authors evaluate the proposed SpodNet on synthetic data and the animal dataset against GLAD, GLasso, and other traditional approaches.

**Strengths:**

The paper is well-written and easy to follow. The idea of unrolling the column-row BCD algorithm to ensure SPD seems novel.

**Weaknesses:**

I think this is a borderline paper in its current form. I value the novelty of the paper, but its numerical performance is not the most convincing. I think solving the following points will make the paper stand more firmly at my rating.

1. GLAD-Z: SpodNet's NMSE performance on synthetic data seems to be consistently worse than GLAD-Z's. I understand the authors' argument that GLAD-Z is not SPD, but what if Z is projected onto the SPD cone? Will the projected Zs remain the lower NMSE scores? Because GLAD uses an ADMM-like algorithm, the learned $\Theta$ matrices are not projections if I understand correctly.
2. Large sample regime: The NMSE performance of SpodNet is no better or only marginally better than the baselines when $n>p$.
3. Real-world datasets: The results of GLAD on the animal dataset are missing. Also, the paper will benefit from adding at least another real-world dataset.
4. Figures: Some figures can be hard to read, especially Fig. 4-5. I suggest the authors use thinner lines and/or redesign their layout to make line plots larger.

**Questions:**

1. Training details: What are the exact steps of the unrolling algorithm? How many unrollings are needed? From line 363, does the training posit an MSE loss on the intermediate $\Theta$s or only the last one?
2. GLasso: In Fig. 5, the F-1 performance of GLasso decreases when $n$ gets larger. This is weird because I expect it to recover the graph perfectly when $n\gg p$. What are the authors' explanations about this?

---

> ### Author Response · Authors · 2024-11-20
> **Response to MbyP**
>
> Dear reviewer,
>
> Thank you for your comments and interesting questions, we are happy that you enjoyed reading the paper. Here is our response:
>
> 1. You are correct: GLAD-Z is not the projection of GLAD-$\Theta$, nor the other way around. **GLAD-Z loses its sparsity if projected onto the SPD cone** and additionally becomes singular (since it is actually projected onto the closed cone of semi-definite matrices), **see the added Figure 10 (line 802)**. Going the other way around and trying to sparsify GLAD-$\Theta$ instead, by thresholding its off-diagonal entries, breaks its positive-definiteness guarantee [1]. To the best of our knowledge, there is no obvious way to modify GLAD or its predictions such that a single matrix guarantees **both** sparsity and positive-definiteness. **We emphasize that GLAD-Z’s outperformance comes at the price of the arguably prohibitive sacrifice of the SPD guarantee, which makes its consideration as a valid precision matrix questionable.**
> 2. We agree that SpodNet does not significantly outperform the baselines in the easy large-sample regimes ($n \gg p$, where all traditional methods already achieve excellent performance (down to 2% NMSE in the $p=20$ scenario)). However we emphasize that **many real-world applications require methods that work well in limited-sample regimes ($n \approx p$ to $n < p$), which is the most difficult setting.** This happens for instance in many time-series applications where signals are assumed to be strongly non-stationary, such as in dynamic connectivity estimation in neuroscience [2, 3, 4] or temporal financial networks [5, 6].
> 3. **We have added the results of GLAD on the Animals dataset (see the updated Figure 6, line 512) and modified Section 5.2 accordingly.**
> 4. We increased the sizes of Figures 4 and 5.
> 5. The models used in our experiments comprise one SpodNet layer, as addressed in Appendix A.2. We recall that a SpodNet layer updates **all** column-row pairs and diagonal elements of the input, thus performing much more than one single update. Since a single SpodNet layer is enough in our experiments to yield competitive performance against the baselines with all three of our models, we did not pursue architectural search with more unrolled layers. In line 363, the MSE therefore only involves the final $\Theta$ output.
> 6. This is an interesting fact indeed. First, notice that the MSE indeed goes down when $n$ increases. The F1 score does not improve because of the choice of $\lambda$ which is tuned by Cross Validation *on the left out MSE score*, as done by the standard scikit-learn’s `GraphicalLassoCV` solver that we used ([7]). It is well-known in the sparsity literature that the optimal regularization strength $\lambda$ in terms of MSE is usually lower than the optimal one for F1 score (see eg.  in the Lasso case, Figure 1 in [8]). Since the F1 score cannot be computed in real life, practitioners resort to MSE on left-out data to tune $\lambda$. This may yield a suboptimal value in terms of support recovery performance.
>
> _Refs_:
>
> [1]: Functions preserving positive definiteness for sparse matrices. Guillot & Rajaratnam, 2015
>
> [2]: Defining epileptogenic networks: contribution of SEEG and signal analysis. Bartolomei et al., 2017.
>
> [3]: Brain functional and effective connectivity based on EEG recordings: a review. Cao et al., 2021.
>
> [4]: Automatic detection of epileptic seizure events using the time-frequency features and machine learning. Zeng et al., 2021.
>
> [5]: Non-Stationarity in financial time series and generic features. Schmitt et al., 2013.
>
> [6]: The physics of financial networks. Bardoscia et al., 2021.
>
> [7]: https://scikit-learn.org/dev/modules/generated/sklearn.covariance.GraphicalLassoCV.html
>
> [8]: Beyond L1: Faster and Better Sparse Models with skglm. Bertrand et al., 2022.

---

> > ### Comment · Reviewer_MbyP · 2024-11-22
> > **Response to the Rebuttal**
> >
> > I thank the authors for their detailed reply. I think showing the exact weighted graphs in Fig. 9 (which the authors referred to as Fig. 10) might be more helpful. My other concerns are resolved more or less. I am happy to keep my score.

---

> > > ### Author Response · Authors · 2024-11-27
> > >
> > > Dear reviewer,
> > >
> > > We are glad to read that we were able to address your concerns ! For completeness, we have added the graphs represented by the matrices in Fig. 9 (line 802)as you suggested, but since GLAD-$\Theta$ and the projected GLAD-$Z$ are non-sparse, they result in fully dense graphs that are difficult to interpret visually, which is why we initially illustrated our response through the support of the matrices. We thank you again for your valuable inputs, and hope to have adequately addressed your last suggestion !

---

### Official Review · Reviewer_MngH · 2024-11-04

**Soundness:** 3
**Presentation:** 4
**Contribution:** 3
**Rating:** 8
**Confidence:** 4

**Summary:**

The paper proposes a new method SPODNET for learning SPD matrices by elements, supported by the classical Shur’s condition, where the matrix elements (u and v) to update are learned using neural networks. The work demonstrates the use of SPODNET for the sparse precision matrix learning task, and proposes three new model architectures to perform the learning, including UBG, PNP and E2E. Two sets of experiments were conducted for evaluation, using a synthetic data and a real-world data. The results show the effectiveness of the proposed methods, and their advantages over the compared ones.

**Strengths:**

•	To my knowledge, the proposed method is original. I like the proposed SPODNET, building on classical theory, simple but elegant.
•	The paper is well written and well structured.
•	Experiment design is appropriate for demonstrating the effectiveness of the proposed method. I particularly like the result of UBG in Figure 6, highlighting more distinct structure that is interesting  for this real-world data.

**Weaknesses:**

•	The paper would be stronger if they could include another real-world learning problem over SPD manifold.  But I don’t see this as a major issue.
•	Lack of discussion on the limitation of the proposed work.
•	There are a couple of things that could be explained better, see my questions.
•	Figures 4 and 5 are too small, hard to read.

**Questions:**

•	Although the proof is straightforward, it would be useful for the reader and for completeness to explain how Eq. (3) is derived in proof.
•	For clarify, the role/design rationale behind each term in Eq. (5) can be explained briefly, although it is an existing method.
•	In line 364, it is mentioned that the MSE reconstruction loss is used. How is this implemented together with the GLasso loss in Eq. (5)?
•	How does the proposed method perform in terms of training time/cost?
•	What do different colours mean in Figure 1?

---

> ### Author Response · Authors · 2024-11-20
> **Response to MngH**
>
> Dear reviewer,
>
> Thank you for your comments and interesting questions, we are happy that you enjoyed reading the paper. Here is our response:
>
> - **We have added a discussion on the current limitations of SpodNet in our conclusion (lines 525-528).** Namely, we would like to improve our theoretical understanding of the eigenvalues’ dynamics during training, and make the training faster.
> - We have increased the sizes of our figures.
> - Eq. (3) stems from the Banachiewicz inversion formula (stated under its general form in Eq. (4)); to obtain Eq. (3) we namely identify $g(\Theta)$ with $\theta_{22} - \theta_{21} [\Theta_{11}]^{-1} \theta_{12}$. **This has been clarified in the text (lines 195-196).**
> - **We have added explanations on Eq. (5), the GLasso loss (lines 233-235).** The data fidelity term is the negative log-likelihood under Gaussian assumption, while the penalty enforces sparsity of $\Theta$.
> - The MSE is used only to train the data-driven models (SpodNet and GLAD). The GLasso estimator is obtained by solving Eq. (5) and its hyperparameter $\lambda$ tuned by cross-validation, using Scikit-Learn’s `GraphicalLassoCV`. Then the MSE is measured for the solution obtained by the GLasso.
> - The SpodNet layer performs block-coordinate updates on the input matrix, each of which in $\mathcal{O}(p^2)$, resulting in an $\mathcal{O}(p^3)$ complexity for a complete update, as mentioned in the paper (line 200). This cost is the same as an update of $\Theta$ for many solvers, e.g. Graphical ISTA or Graphical Lasso, and makes inference quite fast (less than 1 s, which is similar to solving one GLasso). As always with learning-based methods, the cost of training on many examples is more important, in the order of hours ; future directions involve making training faster.
> - Colors in Figure 1 are an illustrative choice. At a given iteration inside a SpodNet layer, a new column is predicted by a neural network. This new column is represented by a colored block. Colors were used to emphasize that each column is different, with no particular meaning behind each color choice. The green color only represents the input matrix.

---

> ### Comment · Reviewer_MngH · 2024-11-25
>
> I thank the authors for the response. They have answered my questions, and I am happy to maintain my positive rating.

---

> > ### Author Response · Authors · 2024-11-27
> >
> > Dear reviewer,
> >
> > We are glad to read this sentiment. Thank you for your valuable inputs !

---

### Author Response · Authors · 2024-11-20
**Global response**

We thank you all for reviewing our work and your positive feedbacks. We are glad that the novelty of our proposed framework is appreciated (MngH, MbyP, R2Se) and that its effectiveness in the conducted experiments is highlighted (MngH, R2Se). We are also pleased that our presentation was clear (MngH, MbyP, R2Se) !

We have provided individual responses to each of you. You may also find attached our updated paper that was modified according to your suggestions and remarks (changes are in blue). We hope that our answers and arguments to your interrogations are helpful, and remain at your service for further ones you may have !

---

### Meta-Review · Area_Chair_g4ad · 2024-12-21

**Metareview:**

The authors of this paper propose a method for constraining neural network outputs (or intermediate representations) to be positive definite and sparse, which can be of interest to a variety of applications (for example, learning a graphical affinity matrix).  This is done by deriving an update to one row/column of the matrix which is guaranteed to preserve the positive definiteness of the overall matrix and which also directly parameterizes the off-diagonal entries of the row/column which allows for the imposition of the sparsity constraints.

The reviewers are generally positive and appreciate the novel contribution of the paper.  Many of the weaknesses noted by the reviewers would be appear to be relatively minor and/or clarification questions and should be addressable in preparing a final version of the manuscript.

**Additional Comments On Reviewer Discussion:**

The authors were responsive in addressing the concerns and questions from the reviewers.

---

### Decision · Program_Chairs · 2025-01-22

Accept (Poster)